# Protein Enrichment of Donor Breast Milk and Impact on Growth in Very Low Birth Weight Infants

**DOI:** 10.3390/nu13082869

**Published:** 2021-08-20

**Authors:** Ting Ting Fu, Heather C. Kaplan, Trayce Fields, Alonzo T. Folger, Katelyn Gordon, Brenda B. Poindexter

**Affiliations:** 1Perinatal Institute, Division of Neonatology, Cincinnati Children’s Hospital Medical Center, Cincinnati, OH 45229, USA; heather.kaplan@cchmc.org (H.C.K.); brenda.poindexter@emory.edu (B.B.P.); 2Department of Pediatrics, University of Cincinnati College of Medicine, Cincinnati, OH 45267, USA; alonzo.folger@cchmc.org; 3TriHealth Good Samaritan Hospital, Cincinnati, OH 45220, USA; trayce_fields@trihealth.com; 4Division of Biostatistics and Epidemiology, Cincinnati Children’s Hospital Medical Center, Cincinnati, OH 45229, USA; 5Department of Pediatrics, Cincinnati Children’s Hospital Medical Center, Cincinnati, OH 45229, USA; katelyn.gordon@cchmc.org; 6Department of Pediatrics, Emory University, Atlanta, GA 30322, USA; 7Division of Neonatology, Children’s Healthcare of Atlanta, Atlanta, GA 30322, USA

**Keywords:** donor breast milk, human milk, milk analysis, very low birth weight, preterm, growth

## Abstract

Protein content is often inadequate in donor breast milk (DBM), resulting in poor growth. The use of protein-enriched target-pooled DBM (DBM+) has not been examined. We compared three cohorts of very low birth weight (VLBW) infants, born ≤ 1500 g: DBM cohort receiving > 1-week target-pooled DBM (20 kcal/oz), MBM cohort receiving ≤ 1-week DBM, and DBM+ cohort receiving > 1-week DBM+. Infants followed a standardized feeding regimen with additional fortification per clinical discretion. Growth velocities and z-scores were calculated for the first 4 weeks (*n* = 69 for DBM, 71 for MBM, 70 for DBM+) and at 36 weeks post-menstrual age (*n* = 58, 64, 59, respectively). In total, 60.8% MBM infants received fortification >24 kcal/oz in the first 30 days vs. 78.3% DBM and 77.1% DBM+. Adjusting for SGA, length velocity was greater with DBM+ than DBM in week 1. Average weight velocity and z-score change were improved with MBM compared to DBM and DBM+, but length z-score decreased similarly across all groups. Incidences of NEC and feeding intolerance were unchanged between eras. Thus, baseline protein enrichment appears safe in stable VLBW infants. Weight gain is greatest with MBM. Linear growth comparable to MBM is achievable with DBM+, though the overall length trajectory remains suboptimal.

## 1. Introduction

Donor breast milk (DBM) is recommended as an alternative to preterm formula for supplementing maternal breast milk (MBM) in very low birth weight (VLBW) infants, but nutritional concerns remain [1]. Some human milk banks utilize target-pooling to combine milk of multiple donors strategically, rather than randomly, to reduce nutritional variability and potentially increase nutritional content, commonly attempting to achieve a minimum caloric density of 20 kcal/oz (67 kcal/dL) [2]. However, even with this technique, protein concentration is often inadequate [3,4]. Furthermore, VLBW infants fed calorically targeted DBM still demonstrate growth faltering, particularly in linear growth [3]. While early protein intake and linear growth have both been associated with neurodevelopmental outcomes [5,6,7], enteral provision of protein via human milk is typically limited by baseline protein content, which is notably lower in DBM compared to MBM. This early enteral protein deficit is additionally compounded by the time required to reaching a feeding volume at which multicomponent fortification can occur.

A growing number of studies have examined the use of targeted fortification in the preterm or VLBW population by analyzing human milk composition and adjusting macronutrient content to meet goal concentrations. One recent randomized controlled trial showed improved postnatal growth with targeted intake to meet recommendations from the European Society of Pediatric Gastroenterology, Hepatology, and Nutrition (ESPGHAN), including 4.5 g/kg/day protein [8,9]. Other smaller studies that targeted protein concentrations only or macronutrient concentrations that did not meet ESPGHAN intake or protein:energy ratio recommendations found more mixed results [10,11,12,13,14,15,16,17,18]. Although targeted fortification may be feasible, barriers to widespread use exist. The process is laborious and time-intensive and requires local access to a human milk analyzer, which may be cost-prohibitive. Additional challenges specific to point-of-care human milk analysis include accurate machine calibration and reliable operator performance [19].

We previously reported that, despite target pooling, the average protein concentration in DBM remained 0.9 g/dL, consistent with mature milk of mothers of term infants rather than milk of preterm mothers from the first month of life [3]. Given that the target-pooled DBM utilized by our unit already aims for a minimum goal of 20 kcal/oz, we next wished to improve early protein delivery by adding a single-component fortifier. By empirically increasing the baseline protein concentration in target-pooled DBM, we sought to achieve an energy and protein profile closer to preterm MBM without utilizing human milk analysis. Thus, the purpose of this study is to describe the use of protein-enriched target-pooled DBM (DBM+) in VLBW infants and compare their growth patterns with those of infants receiving standard target-pooled DBM and MBM.

## 2. Materials and Methods

This observational cohort study was performed at the neonatal intensive care unit (NICU) at TriHealth Good Samaritan Hospital in Cincinnati, Ohio, and was approved by the Institutional Review Board with waiver of informed consent. DBM+ was implemented clinically in January 2019. We prospectively collected data on the outcomes of babies receiving DBM for our previous study [3] and those receiving DBM+ for this study; data for the MBM cohort were collected retrospectively.

### 2.1. Study Population

The DBM cohort consisted of 85 eligible VLBW infants (born weighing less than or equal to 1500 g) admitted to the NICU from December 2015 to April 2017 who received more than 1 week of DBM as previously described [3]. We identified a same size cohort of infants over a similar timeframe (December 2015 to June 2017) who had MBM well-established by 7 days, with less than or equal to 1 week of DBM use (MBM cohort). We then identified a third size-matched cohort consisting of eligible infants admitted to the NICU between January 2019 and August 2020, reflecting the period after DBM+ was implemented (DBM+ cohort). In the DBM+ era, those who received DBM instead of DBM+ were excluded. For all groups, infants were excluded if they transferred to another hospital or passed away in the first 30 days of life, did not follow the standardized feeding protocol (see below), did not receive any donor milk due to parent refusal or clinical discretion, or had an abbreviated donor milk eligibility window due to donor milk shortage or early ad lib feeding (i.e., not due to growth concerns).

### 2.2. Standardized Feeding Approach

Per the unit’s standardized feeding protocol, parenteral nutrition is initiated on admission, providing 2.5 g/kg/day of protein the first day, then 3.5 g/kg/day onward until intake is limited by fluid volume. Donor milk obtained from the OhioHealth Mothers’ Milk Bank (OMMB) is utilized for the first 30 days of life for VLBW infants when maternal milk is not available; after 30 days, infants are transitioned to formula if supplementation is needed. Per OMMB, at the time of this study, donor milk is target-pooled to achieve a minimum of 20 kcal/oz based on analysis from a near-infrared human milk analyzer. Enteral feedings are initiated within 48 h of birth at 15 mL/kg/day for 3 days and subsequently advanced by 10 mL/kg/day every 12 h to a goal of 160 mL/kg/day. Fortification to 24 kcal/oz (81 kcal/dL) occurs at 75 mL/kg/day, approximately day of life 7, using Similac (Abbott Nutrition, Chicago, IL, USA) Human Milk Fortifier Hydrolyzed Protein Concentrated Liquid (HMF). Additional fortification occurs as clinically indicated for suboptimal growth using Similac HMF, Similac Special Care 30, and/or Similac NeoSure, per dietitian’s discretion. Of note, use of Abbott products is standard for the unit.

During the first cohort era (DBM and MBM cohorts), Similac Liquid Protein Fortifier (LPF), an extensively hydrolyzed protein product, was used as needed to provide additional protein if linear growth was suboptimal with multicomponent fortification alone. During the second cohort era (DBM+ cohort), 6 mL of LPF was routinely added to 90 mL of target-pooled DBM to yield DBM+, and DBM+ was utilized with the initiation of enteral feeds. Assuming that the baseline protein concentration of DBM is 0.9 g/dL and energy is 20 kcal/oz, the addition of LPF raises the protein concentration to 1.9 g/dL (Table 1). Fortification to 24 kcal/oz with HMF yields 3.2 g/dL protein and 23.4 kcal/oz. With an enteral intake of 150–160 mL/kg/day, this provides 4.7–5.1 g/kg/day protein, 118.4–126.3 kcal/kg/day, and a protein:energy ratio 4 g/100 kcal.

### 2.3. Clinical Data Collection and Outcomes

Nutritional data, including feeding volumes, fortification, and duration of donor milk intake, were recorded from the medical chart. The percentage of donor milk intake was calculated by dividing the donor milk intake volume by the total human milk volume in the first 30 days.

To assess safety outcomes, feeding intolerance (defined as NPO status due to abdominal concerns longer than 48 h) and clinical diagnoses of necrotizing enterocolitis (NEC) were tracked. For the 2015–2017 era, feeding intolerance and NEC data were reported from the initial cohort of 235 screened infants. Because there were not enough infants eligible for the MBM group in the initial 235 patients, an extra 32 VLBW infants needed to be screened. Some of the 32 patients received more than 1 week of DBM and were ineligible for the MBM cohort, but since the DBM group was already complete, detailed clinical information could not be collected for them as it would have exceeded the number of patients approved by the Institutional Review Board. Thus, it was impossible to report the incidence of feeding intolerance and NEC for all 267 screened patients from 2015 to 2017.

Weight, length, and head circumference (HC) measurements obtained for routine clinical care were collected weekly from birth until 4 weeks of age and at 36 weeks post-menstrual age (PMA). Because length and HC are measured on Sundays in our unit, the week 1 time point was defined as age 7–13 days. Growth velocities, Olsen body mass index (BMI) [20], and Fenton z-scores [21] were calculated for each time point. Weight velocity was calculated using the average two-point model. Average velocities over the four-week period were calculated using measurements at birth and at week 4. In the DBM and MBM era, per unit practice, length boards were used as needed to verify measurements; in the DBM+ era, recumbent length boards were utilized as standard of care. Outliers in length and HC were excluded. For patients discharged prior to 36 weeks PMA, measurements at 35 weeks PMA were recorded if available. Small for gestational age (SGA) was defined as a birth weight below the 10th percentile.

### 2.4. Statistical Analysis

Analyses were performed using SAS Studio version 3.8 software (SAS Institute Inc., Cary, NC, USA). *p* < 0.05 was considered statistically significant. Infant characteristics were compared across cohorts with one-way analysis of variance with Tukey–Kramer adjustment for post hoc multiple comparisons for parametric data, Kruskal–Wallis and Mann–Whitney U tests for non-parametric data, and χ^2^ tests for categorical variables.

Weekly and average growth velocities and z-score changes from birth to 36 weeks PMA were analyzed across groups first by ANOVA with Tukey-Kramer adjustment. To control for covariates, generalized linear modeling estimated by maximum likelihood was used for growth velocity outcomes at each time point with the cohort group representing an independent class. While both SGA status and birth weight were different across groups, SGA was selected as a covariate due to the potential for distinctive growth patterns in this population. Duration or percentage of donor milk intake were not included as covariates since the degree of donor milk exposure was intrinsic to defining each cohort. Fortification of human milk beyond 24 kcal/oz was also not included in the model since the decision to increase the caloric density was typically a response to suboptimal growth and dependent on the outcome itself.

For z-score trajectories, longitudinal analysis using a generalized linear model with repeated measures was utilized, and the interaction between cohort group and age at the time of each measurement was examined. Both an unadjusted model and a model adjusting for SGA were performed for each growth parameter.

Patients were excluded from growth velocity analysis if they had incomplete growth measurements in the first 4 weeks or had modified feeding plans (e.g., prolonged NPO status, continuous feedings, atypical formula exposure). Additional patients were excluded from z-score and longitudinal analysis if they were discharged prior to 35 weeks PMA or had modified feeding plans after the first 30 days. No patients were excluded due to poor growth.

## 3. Results

Figure 1 depicts the flow diagram for infants included in all three cohorts. In total, 267 infants were screened for the MBM cohort, and 270 were screened for the DBM+ cohort. Per the clinical team’s discretion, 27 infants received unenriched DBM instead of DBM+ as a backup to MBM: 19 infants due to critical illness or gestational age (GA) of 22–24 weeks, though one 24-week infant was included in the cohort, and eight other infants for a variety of reasons (Appendix A). A total of 69 infants from the DBM cohort, 71 infants from the MBM cohort, and 70 infants from the DBM+ cohort had 4 weeks of adequate growth data. Of these, 58 infants in the DBM cohort, 64 infants in the MBM cohort, and 59 infants in the DBM+ cohort had measurements available at 36 weeks PMA.

Infant characteristics by cohort are found in Table 2. Although GA was similar in all three groups, the DBM cohort had a lower mean birth weight and a greater proportion of SGA infants compared to DBM+ (*p* = 0.04 and 0.05). Median GA of SGA infants in each group were 29.9 (IQR 28.9–31.6, *n* = 11) for DBM, 28.5 (IQR 26.0–30.7, *n* = 6) for MBM, and 32.1 (IQR 31.2–32.6, *n* = 4) for DBM+. More infants in the DBM and DBM+ cohorts (78.3% and 77.1%) received fortification beyond 24 kcal/oz in the first 30 days compared to the MBM cohort (60.8%, *p =* 0.02 and 0.03). There was a slightly higher percentage of donor milk intake in the DBM+ cohort compared to DBM but no difference in the number of infants who transitioned off donor milk prior to 30 days due to growth concerns.

Incidence of NEC was unchanged between the two eras, as was the proportion of infants receiving MBM at the time of diagnosis (*p* = 0.85) (Table 3). Though not statistically significant, a smaller proportion of infants was receiving donor milk of either type in the DBM+ era (23.5% vs. 50% in DBM era, *p* = 0.11), and a higher proportion was receiving formula (29.4% vs. 6.3% in DBM era, *p* = 0.08). There was no difference in the rate of feeding intolerance.

Weight velocity was greater with MBM in week 2 and averaged over 4 weeks (Table 4). This was similarly reflected in an attenuated decrease in weight z-score from birth to 36 weeks PMA (Table 5, Figure 2). After adjusting for SGA, these findings remained significant for both weight velocity (Table 6) and weight z-score (Table 7). Furthermore, of the infants who were not SGA at birth, 11/49 (22.4%) were SGA at 36 weeks PMA in the DBM group compared to 4/59 (6.8%) in MBM and 8/55 (14.5%) in DBM+ groups, but there was no statistically significant difference as a whole.

Length z-score decreased by 0.94–1.00 without any intergroup difference (Table 5). In the multivariable models adjusting for SGA (Table 8), length velocity was higher with DBM+ compared to DBM in week 1 (DBM reference 0.59 cm/week, DBM+ estimate difference 0.31 cm/week, 95% CI 0.04 to 0.58) and on average (DBM reference 0.88 cm/week, DBM+ estimate difference 0.082 cm/week, 95% CI −0.0085 to 0.17). Accordingly, the change in BMI z-score was greater with MBM and DBM compared to DBM+ in both the ANOVA and longitudinal models (Table 5, Table 7, and Table 9). Additionally, the change in length z-score over time was not different between groups, though there was a slight difference between the change in HC z-score over time between DBM and DBM+ cohorts (DBM+ estimate difference −0.0046 change in z-score per day, 95% CI −0.0093 to 0.0001) (Table 9, Figure 3 and Figure 4).

## 4. Discussion

Targeted protein fortification in human milk feedings has been examined in previous studies, but to our knowledge, none to date have described the use of baseline protein enrichment to improve early delivery with the initiation of enteral feedings and prior to multicomponent fortification [10,12,14,16,18]. Although multicomponent fortification is necessary to meet recommended nutrient intake, there is no consensus on the optimal timing of fortification in VLBW infants, which is often delayed due to the theoretical risk for NEC and feeding intolerance, despite lack of evidence of an association between early fortification and NEC [22,23,24]. In this study, we demonstrate that baseline enrichment with the addition of extensively hydrolyzed protein can be safe in stable VLBW infants, both prior to and in combination with multicomponent fortification, as suggested by the unchanged incidences of NEC and feeding intolerance. Although the clinical team opted not to use DBM+ in nine infants due to periviable gestational age, the DBM+ cohort did include one 24-week infant. However, while DBM+ may be safe, we did not observe a dramatic difference in growth.

Overall, all cohorts were able to attain adequate growth velocities over time. Nonetheless, in all three groups, the majority of infants received fortification concentrations greater than 24 kcal/oz in order to achieve acceptable velocities by weeks 3–4 (15 g/kg/day for weight, 1 cm/week for length and HC) [25,26]. As expected, the proportion of infants receiving greater than 24 kcal/oz was significantly less with MBM, but interestingly, the percentage was similar between the two donor milk cohorts, indicating that the clinical team still often observed growth faltering in the DBM+ group. Despite attaining recommended growth velocity targets, all groups had a net negative change in z-score across the three primary parameters, supporting growing evidence that the current velocity goals do not support growth at a rate to match intrauterine growth references [27,28,29]. Specifically, prior to 34 weeks, approximately 18 g/kg/day in weight gain and 1.4 cm/week of linear growth is required to follow the Fenton curves [29]. Although it is unknown whether achieving postnatal growth patterns parallel to the reference fetus is feasible for all preterm infants, there remains an important need to establish updated velocity standards and determine ideal growth trajectories to optimize long-term outcomes.

In our DBM+ and MBM cohorts, length velocities and z-score trajectories were comparable. Furthermore, length velocity was greater with DBM+ compared to DBM in week 1, when all infants were receiving unfortified milk, likely reflecting the increased enteral protein intake contributed by baseline enrichment, and trended towards significant when averaged over 4 weeks as well. Given the high projected intake of protein for the DBM+ group, these findings are consistent with studies that have shown a stronger relationship between protein intake and linear growth rather than weight gain [28,30]. However, the net decrease by almost a full length z-score in all cohorts is particularly concerning since length is often regarded as a proxy for lean mass accretion and brain growth, and poor linear growth is associated with worse neurodevelopmental outcomes [31,32]. It is unclear if the similar trajectories over time were a result of the high proportion of fortification in the donor milk groups or inadequate power, though the large standard deviations in change in z-scores (Table 5) would suggest the latter. Moreover, since we did not analyze individual milk samples, our protein delivery still might have been inadequate. While most studies of enteral protein have examined intake of 2.6–4.0 g/kg/day [33], higher protein intake greater than 4.6 g/kg/day can be tolerated and is associated with improved growth [28]. Further research of higher enteral protein provision in this population is warranted.

We also report here that average weight velocity was greater with MBM than both DBM and DBM+, with the greatest disparity seen in week 2, as infants were nearing full enteral feeding volumes and receiving milk fortified to 24 kcal/oz at most. There are a few possible explanations for the lower weight velocity seen with DBM and DBM+. First, there may still be differences in actual fat content. Though the target-pooled donor milk purchased by our unit theoretically contains at least 20 kcal/oz, the pooled milk is analyzed by the milk bank prior to bottling. As breast milk is not a homogenous fluid, this analysis might not accurately reflect the content of each individual bottle. Similarly, a study of donor milk purchased from five different milk banks that provided labels with nutrient content found significant differences in reported and measured content, with only 18.7% of samples measuring within 5% of the labeled content [34]. The accuracy of human milk analyzers and the variation in the conversion factors used to determine caloric content pose additional concerns regarding reliability [19,35]. Second, many properties of human milk are modified by the pasteurization process. The extra freeze-thaw cycles that donor milk is exposed to change the structural integrity of the fat globule, and Holder pasteurization completely inactivates bile salt-stimulated lipase and lipoprotein lipase, impacting fat delivery, digestion, and absorption [36,37,38,39]. Pasteurization may also reduce the bioactivity of growth factors and adipokines and alter metabolic regulation [40,41]. These challenges are highlighted by the study of very preterm infants in which, despite equivalent protein and energy intake after targeted fortification, weight velocity was greater with raw MBM compared to pasteurized MBM and lowest with pasteurized donor milk [11], underscoring the complexity of human milk composition and processing.

Limitations of our study include the observational cohort design as well as the exclusion of the sickest and smallest patients from the DBM+ group. There was a higher proportion of SGA infants in the DBM group, possibly due to a number of SGA infants excluded from the DBM+ group as a result of feeding protocol deviations or donor milk availability, and this was reflected in the lower birth weight in the group. However, we adjusted for SGA status in our regression model and focused on change in z-scores when looking at trajectories over time. Another limitation is the imprecision of length and head circumference measurements, as recumbent length boards were not routinely used in the first era. Furthermore, two feeding practice changes occurred during the first cohort era. First, fortification to 24 kcal/oz was condensed from two steps (22 kcal/oz for 24 h, then 24 kcal/oz the subsequent day) to one step, though this likely did not affect growth significantly since infants were still receiving total parenteral nutrition. Second, the recipe for 26 kcal/oz was modified from utilizing 10 mL of HMF and additional NeoSure powder to 15 mL of HMF, but this changed the overall caloric and protein provision minimally. Lastly, the generalizability of our study is partly limited by the availability of target-pooled donor milk. Although an increasing number of non-profit milk banks are target-pooling, the accuracy of each bank’s milk analysis is unclear.

## 5. Conclusions

Baseline protein enrichment of DBM may be safe in stable VLBW infants. Infants receiving primarily MBM demonstrate greater weight gain compared to both infants fed calorically targeted DBM and those fed DBM+. With the provision of DBM+, linear growth comparable to MBM can be achieved, and there may also be improved early length velocity compared to DBM, though overall length trajectory remains suboptimal for all groups.

## Figures and Tables

**Figure 1 nutrients-13-02869-f001:**
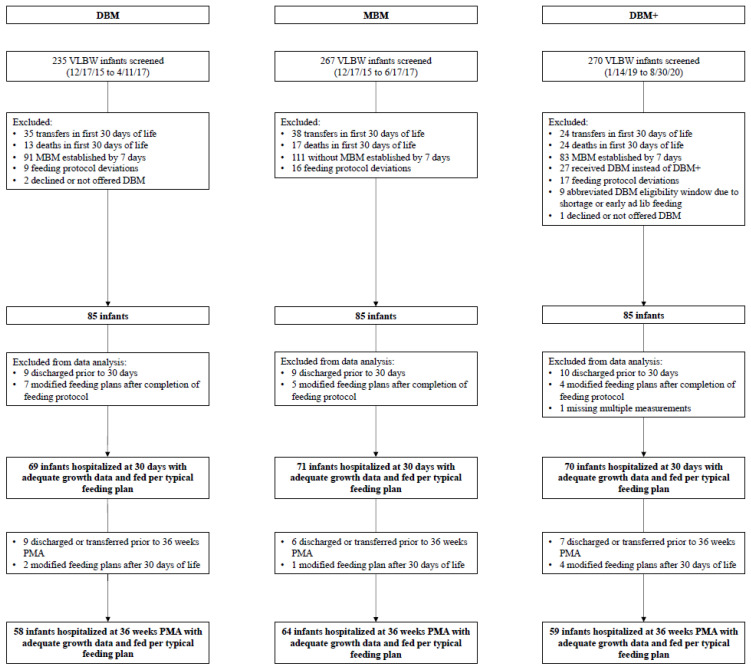
Flow diagram of study infants by cohort. Feeding protocol deviations included delays in advancing volumes, prolonged use of donor milk, and faster advances due to ability to feed orally. Modified feeding plans included continuous delivery, prolonged NPO status of at least 7 days, and atypical formula exposure.

**Figure 2 nutrients-13-02869-f002:**
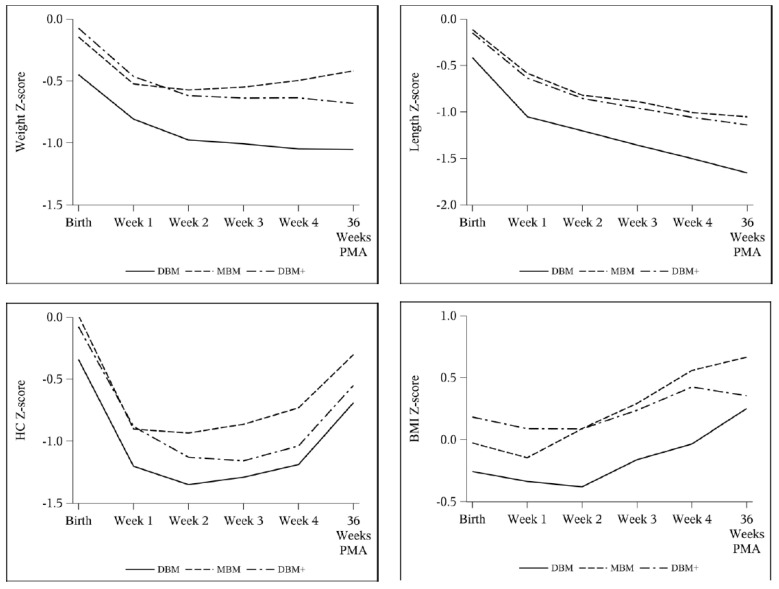
Weekly mean Fenton z-score for weight, length, and HC and Olsen z-score for BMI by cohort: DBM (*n* = 58), MBM (*n* = 64), DBM+ (*n* = 59).

**Figure 3 nutrients-13-02869-f003:**
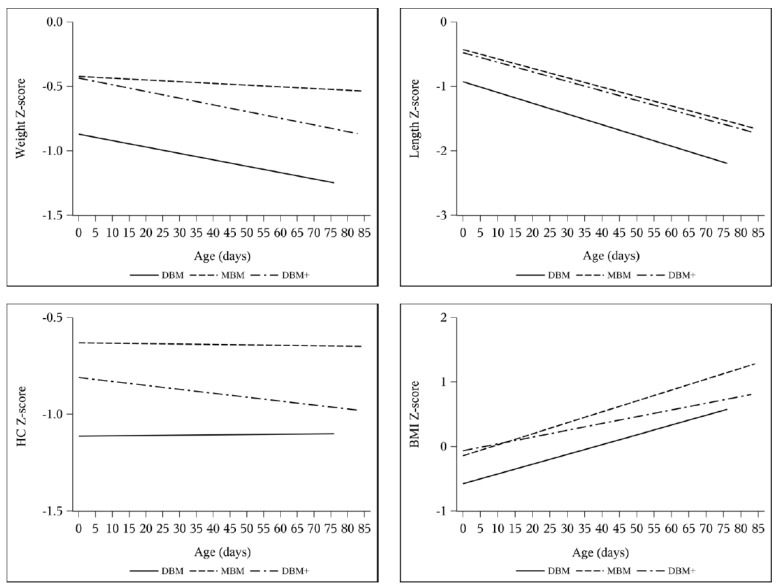
Unadjusted longitudinal models of z-score change over time for Fenton weight, length, and HC and Olsen BMI by cohort: DBM (*n* = 58), MBM (*n* = 64), DBM+ (*n* = 59).

**Figure 4 nutrients-13-02869-f004:**
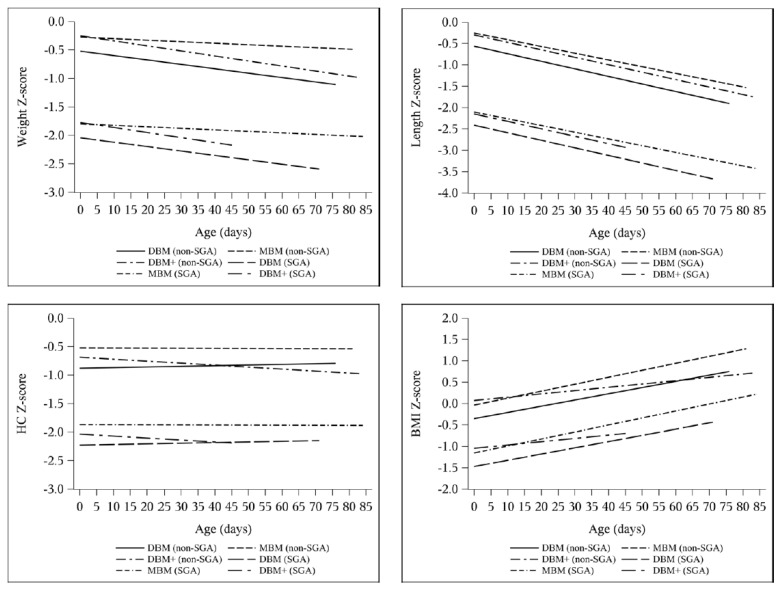
Adjusted longitudinal models of z-score change over time by cohort: non-SGA DBM (*n* = 47), non-SGA MBM (*n* = 59), non-SGA DBM+ (*n* = 55), SGA DBM (*n* = 11), SGA MBM (*n* = 5), SGA DBM+ (*n* = 4).

**Table 1 nutrients-13-02869-t001:** Projected protein and energy concentrations.

	Unfortified		Fortified to 24 kcal/oz	
	Protein (g/dL)	Energy (kcal/oz)	Protein (g/dL)	Energy (kcal/oz)
DBM (purchased from OMMB)	0.9 *	20 *	2.4	23.6
DBM+ (90 mL DBM with 6 mL LPF)	1.9	20.0	3.2	23.4

* Protein content based on previously published analysis [3], energy content based on OMMB analysis. DBM—donor breast milk; LPF—liquid protein fortifier; OMMB—OhioHealth Mothers’ Milk Bank.

**Table 2 nutrients-13-02869-t002:** Infant characteristics by cohort.

	Infants with Growth Data through 30 Days of Life			Infants with Growth Data through 36 Weeks PMA		
	DBM	MBM	DBM+	DBM	MBM	DBM+
	*n* = 69	*n* = 71	*n* = 70	*n* = 58	*n* = 64	*n* = 59
Male (%)	38 (55%)	43 (61%)	32 (46%)	36 (57%)	41 (64%)	26 (44%) *
Gestational age (weeks)	28.9 ± 2.0	28.6 ± 1.8	29.0 ± 1.7	29.0 ± 2.0	28.6 ± 1.8	29.1 ± 1.7
Birth weight (g) ^	1064.4 ± 260.0	1109.0 ± 241.9	1166.3 ± 230.1 **	1053.6 ± 258.5	1108.6 ± 230.9	1169.1 ± 228.6 **
SGA at birth (%)	11 (16%)	6 (8%)	4 (6%) **	11 (19%)	5 (8%)	4 (7%) **
Days on donor milk ^	30 (28–30) *	3 (2–4)	30 (28–30) *	30 (28–30) *	3 (2–4)	30 (29–30) *
Received > 24 kcal/oz in first 30 days (%) ^	54 (78%) *	43 (61%)	54 (77%) *	44 (76%)	40 (63%)	48 (81%) *
% donor milk intake	80.6 (22.8–94.7)		85.0 (47.8–100) **	78.3 (21.9–96.3)		88.7 (43.6–100) **
Transitioned off donor milk early due to growth	5 (7%)		5 (7%)	3 (5%)		4 (7%)

Mean ± SD and ANOVA with Tukey–Kramer adjustment for parametric data. Median (IQR) and Kruskal–Wallis and Mann–Whitney U tests for non-parametric data. χ^2^ tests for categorical variables. ^ *p* < 0.05 for all groups for both infant sets. * *p* < 0.05 compared to MBM; ** *p* ≤ 0.05 compared to DBM. DBM—donor breast milk; DBM+—protein-enriched donor breast milk; MBM—maternal breast milk; PMA—post-menstrual age; SGA—small for gestational age.

**Table 3 nutrients-13-02869-t003:** Incidence of NEC and feeding intolerance between cohort eras.

	2015–2017				2019–2020						*p*-Value
VLBW infants	235				270						
Stage 2 or 3 NEC	14 (6.0%)				15 (5.6%)						0.85
	**MBM**	**DBM**		**Formula**	**MBM**	**DBM+**		**DBM**		**Formula**	
Stage 2 NEC	3	7		1	3	1		0		5	
Stage 3 NEC	2	1		0	3	2		1		0	
Total	7	8		1	8	3		1		5	
	**MBM**		**DBM**		**MBM**		**DBM+**		**DBM**		
Feeding Intolerance	5		4		4		3		1		
**Total**	9 (3.8%)				8 (3.0%)						0.73

NEC staging by Bell’s criteria. Formula exposure occurred only after transitioning off of DBM or DBM+. Feeding intolerance defined as NPO longer than 48 h due to feeding concerns. χ^2^ tests. DBM—donor breast milk; DBM+—protein-enriched donor breast milk; MBM—maternal breast milk; NEC—necrotizing enterocolitis; VLBW—very low birth weight.

**Table 4 nutrients-13-02869-t004:** Group comparisons of weekly growth velocities.

	DBM (*n* = 69)	MBM (*n* = 71)	DBM+ (*n* = 70)	*p*-Value			
				All Groups	DBM vs. MBM	DBM vs. DBM+	MBM vs. DBM+
Weight (g/kg/day)							
Week 1	9.53 ± 7.52	9.07 ± 7.65	9.82 ± 8.07	0.85	0.94	0.97	0.83
Week 2	13.69 ± 6.38	16.70 ± 6.39	13.28 ± 5.72	0.002	0.01	0.92	0.003
Week 3	17.71 ± 5.84	18.84 ± 5.15	17.75 ± 5.19	0.37	0.43	1.00	0.45
Week 4	17.16 ± 5.18	19.20 ± 5.77	17.75 ± 5.74	0.09	0.08	0.81	0.27
Average	13.79 ± 2.97	15.13 ± 2.99	14.03 ± 2.34	0.01	0.01	0.87	0.05
**Length (cm/week)**							
Week 1	0.59 ± 0.85	0.74 ± 0.75	0.90 ± 0.71	0.09	0.55	0.07	0.47
Week 2	0.97 ± 0.72	0.81 ± 0.71	0.81 ± 0.81	0.35	0.42	0.43	1.00
Week 3	0.95 ± 0.78	1.18 ± 0.69	1.10 ± 0.61	0.16	0.15	0.42	0.81
Week 4	1.03 ± 0.64	1.03 ± 0.59	1.08 ± 0.58	0.86	1.00	0.89	0.88
Average	0.90 ± 0.26	0.95 ± 0.23	0.97 ± 0.28	0.24	0.45	0.22	0.89
**HC (cm/week)**							
Week 1	0.14 ± 0.77	0.04 ± 0.81	0.18 ± 0.77	0.58	0.77	0.93	0.56
Week 2	0.67 ± 0.55	0.85 ± 0.55	0.59 ± 0.58	0.03	0.18	0.68	0.03
Week 3	1.01 ± 0.54	1.00 ± 0.51	0.84 ± 0.48	0.10	0.99	0.13	0.17
Week 4	1.01 ± 0.60	1.07 ± 0.58	1.03 ± 0.63	0.83	0.82	0.98	0.92
Average	0.70 ± 0.30	0.74 ± 0.29	0.67 ± 0.27	0.36	0.71	0.80	0.33

Mean ± SD and ANOVA with Tukey–Kramer adjustment. DBM—donor breast milk; DBM+—protein-enriched donor breast milk; HC—head circumference; MBM—maternal breast milk.

**Table 5 nutrients-13-02869-t005:** Net change in growth measurement z-scores from birth to 36 weeks PMA.

	DBM (*n* = 58)	MBM (*n* = 64)	DBM+ (*n* = 59)	*p*-Value			
				All Groups	DBM vs. MBM	DBM vs. DBM+	MBM vs. DBM+
Weight	−0.50 ± 0.47	−0.26 ± 0.51	−0.59 ± 0.40	<0.001	0.01	0.58	<0.001
Length	−1.00 ± 0.57	−0.94 ± 0.55	−0.97 ± 0.50	0.87	0.85	0.95	0.97
HC	−0.22 ± 0.70	−0.30 ± 0.77	−0.46 ± 0.70	0.20	0.84	0.19	0.43
BMI	0.60 ± 0.94	0.74 ± 0.79	0.24 ± 0.72	0.005	0.66	0.06	0.004

Fenton z-scores for weight, length, and HC and Olsen z-score for BMI. Mean ± SD and ANOVA with Tukey–Kramer adjustment. BMI—body mass index; DBM—donor breast milk; DBM+—protein-enriched donor breast milk; HC—head circumference; MBM—maternal breast milk.

**Table 6 nutrients-13-02869-t006:** Linear model of growth velocity comparisons between groups, adjusted for SGA, with MBM as reference.

	MBM (Reference)	DBM Parameter Estimate	*p*-Value	DBM+ Parameter Estimate	*p*-Value
Weight velocity (g/kg/day)					
Average	14.85	−1.59 (−2.46, −0.72)	<0.001	−1.02 (−1.88, −0.15)	0.02
**Length velocity (cm/week)**					
Average	0.94	−0.060 (−0.15, 0.031)	0.20	0.022 (−0.067, 0.11)	0.63
**HC velocity (cm/week)**					
Average	0.72	−0.053 (−0.15, 0.042)	0.27	−0.065 (−0.16, 0.029)	0.18

Generalized linear model with maximum likelihood estimation (95% confidence interval). Growth velocities were the outcomes, and the variables in model were cohort group and SGA status. DBM—donor breast milk; DBM+—protein-enriched donor breast milk; HC—head circumference; MBM—maternal breast milk; SGA—small for gestational age.

**Table 7 nutrients-13-02869-t007:** Longitudinal analysis of z-score change over time, adjusted for SGA, with MBM as reference.

	DBM (vs. MBM) Parameter Estimate	*p*-Value	DBM+ (vs. MBM) Parameter Estimate	*p*-Value
Weight	−0.0051 (−0.0086, −0.0016)	0.004	−0.0062 (−0.0095, −0.0029)	<0.001
Length	−0.0019 (−0.0052, 0.0014)	0.25	−0.0017 (−0.0048, 0.0014)	0.29
HC	0.0013 (−0.0031, 0.0057)	0.57	−0.0033 (−0.0078, 0.0012)	0.15
BMI	−0.0018 (−0.007, 0.0035)	0.51	−0.0086 (−0.0132, −0.004)	<0.001

Fenton z-scores for weight, length, and HC and Olsen z-score for BMI. Generalized linear model with maximum likelihood estimation (95% confidence interval). Growth z-scores were the outcomes, and the variables in model were age in days, cohort group, SGA status, and interaction of age with group. BMI—body mass index; DBM—donor breast milk; DBM+—protein-enriched donor breast milk; HC—head circumference; MBM—maternal breast milk; SGA—small for gestational age.

**Table 8 nutrients-13-02869-t008:** Linear model of growth velocity comparisons between DBM and DBM+ groups, adjusted for SGA.

	DBM (Reference)	DBM+ Parameter Estimate	*p*-Value
Weight velocity (g/kg/day)			
Average	13.27	0.57 (−0.30, 1.45)	0.20
**Length velocity (cm/week)**			
Week 1	0.59	0.31 (0.035, 0.58)	0.03
Average	0.88	0.082 (−0.0085, 0.17)	0.08
**HC velocity (cm/week)**			
Average	0.67	−0.012 (−0.11, 0.083)	0.80

Generalized linear model with maximum likelihood estimation (95% confidence interval) using DBM as reference. Growth velocities were the outcomes, and the variables in model were cohort group and SGA status. DBM—donor breast milk; DBM+—protein-enriched donor breast milk; HC—head circumference; MBM—maternal breast milk; SGA—small for gestational age.

**Table 9 nutrients-13-02869-t009:** Longitudinal analysis of z-score change over time, adjusted for SGA, with DBM cohort as reference.

	DBM+ (vs. DBM) Parameter Estimate	*p*-Value
Weight	−0.0011 (−0.0046, 0.0024)	0.54
Length	0.0003 (−0.0032, 0.0037)	0.88
HC	−0.0046 (−0.0093, 0.0001)	0.05
BMI	−0.0068 (−0.0121, −0.0015)	0.01

Fenton z-scores for weight, length, and HC and Olsen z-score for BMI. Generalized linear model with maximum likelihood estimation (95% confidence interval). Growth z-scores were the outcomes, and the variables in model were age in days, cohort group, SGA status, and interaction of age with group. BMI—body mass index; DBM—donor breast milk; DBM+—protein-enriched donor breast milk; HC—head circumference; MBM—maternal breast milk; SGA—small for gestational age.

## Data Availability

The data presented in this study are available on request from the corresponding author.

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
