# Peer review of "Protein Enrichment of Donor Breast Milk and Impact on Growth in Very Low Birth Weight Infants"

_nutrients, 2021, doi:10.3390/nu13082869_

Round 1

Reviewer 1 Report

Ting Ting Fu et al have conducted a useful study by comparing the effects on growth of 3 different types of feeding in VLBW infants during a short period of life immediately after birth. They found that by administering  protein-enriched target-pooled  donor breast milk (DBM +) vs DBM vs breast milk of the mother did not increased the adverse effects ie NEC but did not improved growth considerably. Although the study is of observational nature and the groups are not optimally matched such information is useful for the neonatologist who strives to find the optimal way to feed the VLBW neonate. The discussion should be more constructive not repeating the results of the study while the limitations of the study are satisfactorily addressed by the authors.

Author Response

Thank you for your thoughtful review of our manuscript. We appreciate your comment regarding the discussion. Because of the amount of data presented in the results, we wanted to provide commentary specific to each important finding in addition to a discussion of its relevance in the context of existing literature. We feel that the reiteration of the most salient results contributes to the clarity of our discussion and have left it unchanged.

Reviewer 2 Report

This observational cohort study compared growth outcomes amongst VLBW infants fed with maternal breastmilk, donor pooled breastmilk and protein-enriched donor pooled breastmilk.  This research has direct clinical relevance; as donor milk becomes more widely available to the VLBW population, the question of how best to promote growth whilst using donor milk is a pressing one.  This is generally a well-written and scientifically sound manuscript.  There are a few points that should be addressed:

  1. Energy intakes are described per oz, whilst all other macronutrients are described per Litre.  It would be helpful to have energy per Litre as well.
  2.  Weight velocity has been calculated using an average 2-point method, which is more susceptible to differences in birthweight than an exponential method (Patel et al, Pediatrics 2005).  Might the differences in early weight velocity be due to the underlying differences in birthweight between groups?
  3. The layout of table 3 is hard to follow, and the inclusion of a "formula" arm without a denominator renders the breakdown of NEC staging numbers fairly meaningless.  Given the study was not powered to find a difference in NEC incidence, this table is more detailed than is required.  The reporting of the NEC rates (Bell stage >=2) between cohorts would probably suffice.
  4. Figures 2, 3 and 4 are small and quite difficult to read- these should be enlarged.  It may be clearer to display MBM as the solid (referent) line.
  5. In the linear models presented in figure 3, the DBM line finishes 10 days before the others.  Please review.
  6. The SGA infants modelled in figure 4 have incomplete data for the full 85 day period, potentially reflecting a more mature gestation.  The definition of SGA used in the paper needs to be included in the methods section, and it might be useful to the reader to have more detail on the demographics of the SGA infants.   
  7.  A comment is made in the discussion on the potential adverse neurodevelopmental consequences of poor linear growth- please also comment on the potential long-term metabolic risks of early discordant growth patterns.

Author Response

Thank you for your thoughtful review of our manuscript. Our responses to your comments are as follows:

  1. We understand the concern with the lack of uniformity of the units and recognize that kcal/oz is standard only in the US. Interestingly, the macronutrient concentrations are still reported in g/dL in the US. We felt that converting every energy value to kcal/dL would be too cumbersome from a readability standpoint; thus, we have added the conversion only to the first mentions of 20 and 24 kcal/oz for reference. If the editors wish for us to provide the calculation for every energy value, please let us know and we will do so accordingly.
  2. We appreciate the valid point regarding the accuracy of the exponential model. However, the mean birth weight was only significantly different between the DBM+ and DBM groups, and the difference in weight velocity was not between DBM+/DBM but rather between DBM/MBM and DBM+/MBM. Furthermore, looking at the Patel reference (DOI: 10.1542/peds.2004-1699 to be consistent), with the average 2-point method, 100% of the infants were <5% different from the accurate standard for both weekly and monthly calculations, similar to the exponential model. “For hospital stays of >160 days, the error exceeded 20% for the 2-point average weight model... This model would thus be acceptable to use for the majority of ELBW infants, because recent studies reported mean LOS for this population of 86 and 100 days and median LOS of 87.5 days… Researchers and clinicians should take care to use this model only for infants with LOS of <160 days.” While we did not directly report the length of stay, as displayed in Figure 4, none of our groups had any measurements past 85 days.
  3. The “All NEC” and “Stage 1” rows have been removed from the table. To improve readability of the layout, a faint gray line has been placed between columns, though this alters the standard table template of the journal; we hope the editors will allow this change to remain. An explanation of the formula exposure has been added to the footnote.
  4. Figures 2-4 have been increased in size.
  5. The figure is correct. The last time point captured was day 76 for DBM, 84 for MBM, 83 for DBM+.
  6. SGA definition is now in the Methods, section 2.3. The median (IQR) GA for the SGA infants in each group has been added to the text of the Results. Given the small proportion of SGA infants in each group, it did not seem meaningful to produce a separate table for SGA infant demographics, but please let us know if you think additional information would be useful.
  7. To our knowledge, currently there are no long-term studies linking early discrepant growth (measured by weight/length ratio, BMI, or body composition) in the preterm population from birth to term corrected age with later metabolic outcomes. While preterm birth and intrauterine growth restriction (or SGA status) are known predisposing factors to metabolic risks later in life and growth patterns later in infancy/childhood have been associated, no relationship with postnatal/neonatal growth alone has been established (Lapillonne and Griffin, 2013, DOI: 10.1016/j.jpeds.2012.11.048). More recent studies examining the relationship of growth and metabolic risks in preterm infants have either (1) included a high proportion of SGA infants, (2) did not look specifically at birth to term corrected age time period, or (3) focused on weight change alone. Furthermore, standardized curves for BMI and air plethysmography body composition in the preterm population were published only in 2015 and 2019 respectively.